# Peptide Adjuvant to Invigorate Cytolytic Activity of NK Cells in an Obese Mouse Cancer Model

**DOI:** 10.3390/pharmaceutics13081279

**Published:** 2021-08-17

**Authors:** Seungmin Han, Minjin Jung, Angela S. Kim, Daniel Y. Lee, Byung-Hyun Cha, Charles W. Putnam, Kwang Suk Lim, David A. Bull, Young-Wook Won

**Affiliations:** 1Division of Cardiothoracic Surgery, Department of Surgery, University of Arizona College of Medicine—Tucson, Tucson, AZ 85724, USA; seungminhan@arizona.edu (S.H.); minjinjung@arizona.edu (M.J.); ask1@arizona.edu (A.S.K.); dylee85@arizona.edu (D.Y.L.); bhcha@arizona.edu (B.-H.C.); cwp@arizona.edu (C.W.P.); davidbull@arizona.edu (D.A.B.); 2Interdisciplinary Program in Biohealth-Machinery Convergence Engineering, Department of Biotechnology and Bioengineering, College of Art, Culture and Engineering, Kangwon National University, Chuncheon 24341, Korea; kslim@kangwon.ac.kr

**Keywords:** prohibitin binding peptide, NK cell, immunotherapy, obesity, tumor microenvironment

## Abstract

Cancer patients who are overweight compared to those with normal body weight have obesity-associated alterations of natural killer (NK) cells, characterized by poor cytotoxicity, slow proliferation, and inadequate anti-cancer activity. Concomitantly, prohibitin overexpressed by cancer cells elevates glucose metabolism, rendering the tumor microenvironment (TME) more tumor-favorable, and leading to malfunction of immune cells present in the TME. These changes cause vicious cycles of tumor growth. Adoptive immunotherapy has emerged as a promising option for cancer patients; however, obesity-related alterations in the TME allow the tumor to bypass immune surveillance and to down-regulate the activity of adoptively transferred NK cells. We hypothesized that inhibiting the prohibitin signaling pathway in an obese model would reduce glucose metabolism of cancer cells, thereby changing the TME to a pro-immune microenvironment and restoring the cytolytic activity of NK cells. Priming tumor cells with an inhibitory the prohibitin-binding peptide (PBP) enhances cytokine secretion and augments the cytolytic activity of adoptively transferred NK cells. NK cells harvested from the PBP-primed tumors exhibit multiple markers associated with the effector function of active NK cells. Our findings suggest that PBP has the potential as an adjuvant to enhance the cytolytic activity of adoptively transferred NK cells in cancer patients with obesity.

## 1. Introduction

More than one-third of the world’s population is overweight; these individuals have an increased risk of cancer because of obesity-associated conditions [1,2,3]. Of particular relevance, obesity causes functional deficiencies in the innate immune system, the first line of defense against oncogenesis. Critical to innate immunity are natural killer (NK) cells, which are specifically weakened in overweight individuals thereby increasing cancer risk [2,4]. As obesity develops, endogenous NK cells transition to a dysfunctional, exhausted state, in which their cytolytic activity against cancer cells is compromised by obesity-induced intracellular lipid accumulation, adipokines, and chronic inflammation [2,3,5,6,7]. Adoptive immunotherapy has therefore been considered a potential treatment option for obese-cancer patients.

However, the tumor microenvironment (TME) in obese patients is also more conducive to cancer cell survival, proliferation, and escape from immune surveillance and is hostile to the activity of immune cells [8,9,10]. A suffice of glucose in the TME, as well as adipokines secreted from adipose tissue, provide energy for cancer cells to aggressively proliferate. The metabolites produced as a consequence of enriched glucose metabolism in cancer cells further alter the state of the TME to a pro-tumor TME one, in which the NK cells’ cytolytic activity is restrained [7,11,12]. The pro-tumor TME, already unfriendly to immune cells, and obesity-initiated high glucose levels further exhaust immune cells, adversely affecting their cytolytic activities; not only endogenous immune cells but also adoptively transferred immune cells are affected [13]. Consequently, priming the pro-tumor TME caused by obesity to a pro-immune microenvironment is thought to enhance innate immune cell-based immunotherapies of cancer in obese patients [12]. We hypothesize that inhibition of the obesity-induced, up-regulated glucose metabolism of cancer cells would remodel the pro-tumor TME and restore the activity of endogenous, as well as adoptively transferred exogenous, NK cells [12,14,15,16].

Overexpression of the receptor prohibitin on the cancer cell membrane favors obesity-associated tumor development and progression [17,18,19,20,21,22,23,24]. Prohibitin interacts with various pathways that upregulate proliferation and enrich glucose metabolism in cancer cells [18,25,26]. Blocking prohibitin pathways would therefore down-regulate the enriched glucose metabolism in cancer cells, favoring conversion from a pro-tumor TME to a pro-immune microenvironment. The prohibitin binding peptide (PBP), CKGGRAKDC, binds to prohibitin expressed on adipocytes as well as cancer cells [27,28,29,30]. PBP is known to bind to both mouse and human prohibitin, which share 99.64% sequence identity. The binding of the peptide to prohibitin inhibits its signaling, in particular via the Ras/C-Raf and Akt/PI3K pathways. The consequential effects upon the cancer cell are reduced glucose metabolism and suppressed tumor growth. In addition, the binding of the peptide to prohibitin in adipocytes decreases their secretion of adipokines; lower levels of adipokines further transform the TME into a state more hostile to cancer cells, and more favorable for immune cells. In the setting of obesity, priming cancers with the PBP peptide to block prohibitin pathways is therefore expected to yield two benefits: (1) suppression of cancer metabolism and (2) conversion of a pro-tumor TME into a pro-immune microenvironment. By employing the peptide as an adjuvant to immune cell-based immunotherapy, we anticipate increased therapeutic efficacy through restoration of the cytolytic activity of NK cells, regardless of their source—exogenous or endogenous.

In this report, we study the peptide as an adjuvant to adoptively transferred NK cell therapy. Initial in vitro studies established that preconditioning with PBP reduced glucose metabolism in cancer cells and enhanced both cytokine secretion and cytolytic activity of NK-92 cells exposed to the PBP-primed cancer cells. To recreate the clinical scenario of obese patients treated with adoptive transfer of NK cells, an obese mouse tumor model was established. Preconditioning mice with PBP improved the therapeutic efficacy of NK cells injected into the tumors, enhancing NK cell cytokine secretion and inhibiting tumor growth, in comparison to mice not receiving the peptide. Notably, adoptively transferred NK cells harvested from the tumors of PBP-preconditioned mice had phenotypic changes indicative of activated effector NK cells. Consequently, PBP shows promise as an adjuvant to adoptive immunotherapy for cancer patients with obesity.

## 2. Materials and Methods

### 2.1. Cells, Culture Media, and Peptide Preparation

The luciferase-expressing B16F10-Fluc-Puro (B16F10) melanoma cell line was purchased from Imanis Life Sciences (Rochester, MN, USA). The cells were cultured according to the manufacturer’s instructions in Dulbecco’s modified essential media (DMEM; Gibco-Thermo Fisher Scientific, Waltham, MA, USA) containing 10% fetal bovine serum (FBS; Gibco-Thermo Fisher Scientific), 1% penicillin/streptomycin (Gibco-Thermo Fisher Scientific), and 1 μg/mL puromycin (InvivoGen, San Diego, CA, USA). NK-92 (human NK, CRL-2407™) cells were obtained from ATCC^®^ (Manassas, VA, USA) and cultured in X-vivo 15 media (Lonza Bioscience, Basel, Switzerland) to which 10% FBS, 1% P/S, and 250 U/mL of interleukin-2 (IL-2; PeproTech, Cranbury, NJ, USA) were added. The peptide CKGGRAKDC (PBP) and GDKCGACRK (PBP scramble; PBP-Scr) were synthesized by Genscript Biotech (Piscataway, NJ, USA). The working solution was prepared in sterile saline and transiently stored at 4 °C.

### 2.2. Assays of In Vitro Cytotoxic Effects of Peptide and NK Cells on Cancer Cells

To measure cancer cell proliferation in response to peptide, B16F10 cells were seeded at a density of 2 × 10^4^ cells/well in 24-well plates 24 h prior to peptide treatment. Various concentrations (0, 1, 10, 50, 100, 150, 200, or 300 μM) of PBP or PBP-Scr were added to the wells and the plates incubated for an additional 4 h or 48 h (*n* = 3 each) at 36 °C with 5% CO_2_ atmosphere. Then, the Cell Counting Kit-8 (CCK-8; Dojindo Molecular Technologies, Inc., Kumamoto, Japan) was used according to the kit instructions to determine the number of viable cells.

To analyze anti-cancer cytolytic activity of NK cells, B16F10 cells were pre-labeled with 25 μM Cell Tracker Blue CMAC (Thermo Fisher Scientific) and seeded at 5 × 10^4^ cells/well in 48-well plates. The following day, 150 μM PBP or PBP-Scr were added to the treatment wells and the cancer cells were primed by incubating for an additional 48 h. At the conclusion of PBP or PBP-Scr priming, the cancer cells were inoculated with NK cells, 150 μM PBP, 150 μM PBP-Scr, or mixtures of NK cells and each peptide and incubated for an additional four hours. NK cells were inoculated at an effector: target (E:T) ratio of 5:1. After the 4 h treatment interval, cells and supernatant were harvested for subsequent analysis to document the anti-cancer activity of NK cells. The cells were labeled with the Annexin V Alexa Fluor 488 and propidium iodide (PI) kit (Thermo Fisher Scientific). Cancer cell death was analyzed in the Cell Tracker Blue-stained population by flow cytometry (BD LSRII; BD Bioscience, Franklin Lakes, NJ, USA). The harvested supernatant was stored at −20 °C for subsequent ELISA assays of the cytokines IFN-_γ_, TNF-α, and granzyme B (R&D Systems, Minneapolis, MN, USA).

### 2.3. Confirmation of Glucose Metabolism Inhibition by PBP

Glucose concentration in B16F10 cell lysates was measured using the Glucose Assay Kit (Abcam, Cambridge, UK). Six-well plates were seeded with 5 × 10^5^ B16F10 cells/well one day before priming with 150 μM PBP or PBP-Scr. After 4 or 48 h of treatment, the cells were harvested and glucose concentrations were assayed following the kit instructions. Fructose concentrations were also analyzed in cancer cells exposed to the PBP peptide. T75 flasks were seeded with 2 × 10^6^ B16F10 cancer cells a day prior to PBP (150 μM) treatment. After a 48 h incubation under standard conditions, the cells were washed three times with DPBS before harvesting. The harvested cell pellets were analyzed for fructose concentration using a fructose assay kit obtained from Abcam, following the manufacturer’s instructions.

### 2.4. Animal Model

Animal procedures (#18-394) were approved in May 2018 by the Institutional Animal Care and Use Committee (IACUC) of the University of Arizona. Eight-week-old male C57BL/6J (stock number: 000664) mice were purchased from Jackson Laboratory (Bar Harbor, ME, USA). To create the obese mouse model, mice were fed with High Fat Diet (HFD, which provides 5.1 kcal/g with 60.3% fat, 18.3% protein, and 21.4% carbohydrate). When the bodyweight reached 40 g, B16F10 cells mixed with Matrigel^®^ matrix (Corning Inc., Corning, NY, USA) were subcutaneously administered. On the same day, mice of normal body weight were also subcutaneously injected with B16F10 cancer cells, as a control to verify that obese mice in fact grow larger tumors. Once their tumors reached 150 mm^3^ in size, the mice were randomly assigned to the various treatment groups.

### 2.5. In Vivo Cancer Model and Measures of Treatment Efficacy

B16F10 mouse melanoma cells (5 × 10^5^ cells per mouse) were injected subcutaneously into the right flank of obese mice, as described above. When the tumor volume reached approximately 150 mm^3^, based on caliper measurements (v = 0.5ab^2^; a = long length, b = short length), the mice were randomly assigned to the following groups (*n* = 8 mice/group): (1) control (PBS); (2) PBP treatment; (3) NK cell treatment; (4) PBP plus NK cell treatment. For adoptive transfer of NK cells into mice of groups 3 and 4, an aliquot of 1 × 10^6^ NK cells in 100 μL PBS was injected directly into the tumor; groups 1 and 2 received 100 μL of PBS only, also directly injected into the tumor. Groups 3 and 4 also received a single intraperitoneal injection of IL-2 (5 × 10^4^ U) on the same day as the NK cell injections. Groups 2 and 4 received PBP, administered twice a week by intraperitoneal injection at a dose of 150 μg/kg in 100 μL PBS, beginning on the day of PBS (Group 2) or NK cell (Group 4) injection into their tumors. 

Bodyweight was measured every two days starting on day 3 and continuing until the conclusion of the study. Tumor size was measured manually on days 3, 5, 7, 9, and 14. Bioluminescence imaging was performed on days 2, 6, and 10; following the intraperitoneal injection of 200 μL (15 mg/mL in PBS) D-luciferin (Gold Biotechnology, Saint Louis, MO, USA), and images were acquired using the Spectral Instruments Imaging LAGO (Spectral Instruments Imaging, Tucson, AZ, USA). Regions of interest (ROI) were defined manually using the image software Aura. After 2 weeks, the mice were sacrificed and the tumors were harvested, snap-frozen, and stored at −80 °C for later analysis of cytokines and surface antigens of NK cells.

### 2.6. Ex Vivo Tissue Analysis: Cytokine and Phenotype of NK Cell

For NK cell-related cytokine analysis, tumor tissues were immersed in liquid nitrogen and physically homogenized before being lysed in RIPA buffer (Thermo Fisher Scientific). The supernatant was collected after centrifugation and IFN-_γ_, TNF-α, and granzyme B concentrations were determined by ELISA, using kits purchased from R&D Systems. Cytokine levels were normalized to total protein concentration, measured with the Pierce™ BCA Protein Assay Kit (Thermo Fisher Scientific).

For NK cell phenotypic analyses, tumor tissues were immersed in Hank’s balanced salt solution (HBSS; Gibco-Thermo Fisher Scientific) containing 2% penicillin/streptomycin, and homogenized, followed by the addition of collagenase IV (Stemcell Technologies, Vancouver, BC, Canada). Subsequently, the cells were washed three times with HBSS, and resuspended in HBSS; aggregates and tissue fragments were then removed by centrifugation. The cells were stained with Anti-CD56-APC, CD16-FITC, and NKG2A-PE antibody (Miltenyi Biotec, Bergisch Gladbach, NRW, Germany), following the instruction manual. Cell samples were analyzed using the BD LSRII and *Flowjo* software (Becton, Dickinson & company (BD), Franklin Lakes, NJ, USA).

### 2.7. Statistical Analysis

Statistical analysis was with Student *t*-test, one-way ANOVA, or two-way ANOVA with Tukey tests using GraphPad Prism 8 software (Graphpad Software, San Diego, CA, USA). Statistical significance thresholds of each test were set at * *p* < 0.03, ** *p* < 0.002, *** *p* < 0.0002, and ns = non-significance.

## 3. Results

### 3.1. PBP Priming Effect on Cancer Cells

In cancer patients who are obese, endogenous NK cells are generally exhausted; consequently, adoptive transfer of NK cells might prove to be an effective treatment of their cancers, especially if coupled with PBP priming of the malignant cells to enhance NK cell activity. To determine a suitable concentration of PBP for in vitro experiments, B16F10 cancer cells were exposed to various concentrations of PBP or a scrambled PBP (PBP-Scr) for either 4 or 48 h, followed by assays of cell viability. Overall, a negative correlation between PBP concentration and cancer cell viability was observed after treatment for 48 h (Figure 1A); PBP-Scr did not have a noticeable effect on B16F10 cell viability (Figure 1B). The effects of PBP on cell survival were however quite modest with viability remaining over 85% at all PBP concentrations evaluated. The lowest concentration to exhibit a demonstrable (less than 90%) reduction of cell viability and to show a significant difference between 4- and 48-h of treatment was 150 μM. Consequently, a concentration of 150 μM coupled with a 48h interval of priming was chosen for the in vitro experiments.

The modest reduction in cancer cell proliferation by PBP, demonstrating that peptide alone was an insufficient treatment of cancer, is likely the consequence of inhibition of prohibitin rather than any manifestation of toxicity [19,25]. If PBP is reducing cancer cell viability via prohibitin inhibition, then a concomitant effect on glucose metabolism is anticipated. To examine this prediction, B16F10 cells were incubated with 150 μM PBP or PBP-Scr for 48 h, at which time cytoplasmic glucose amounts were assayed. As predicted, 150 μM PBP-primed cancer cells had less glucose in their cell lysates than non-treated cancer cells (Figure 1C and Appendix A) while PBP-Scr-primed cancer cells had similar glucose amounts in cell lysates compared to non-treated cancer cells (Figure 1C). Not only did 48 h PBP-primed cells had significantly lower intracellular concentrations of glucose, their TME (the culture medium) also had somewhat higher glucose concentrations than observed with TME from control (non-treated) cells (Appendix A). Taken together, the concentrations of intracellular and extracellular (culture medium) glucose indicate that PBP priming induces inhibition of glucose uptake by cancer cells and thus alters the TME [31,32,33]. Fructose—an intermediate metabolite of glucose—was measured to further probe glucose metabolism. B16F10 cells treated with 150 μM PBP had lower (but not statistically significantly) fructose concentrations than untreated cells (Appendix A), suggesting that blocking prohibitin with PBP in B16F10 cells also inhibited the metabolism of glucose [31]. Taken together, these results imply that PBP-priming inhibits cancer cell proliferation via downregulation of glucose uptake and metabolism.

### 3.2. In Vitro Tumor Priming with PBP Enhances Anti-Cancer Activity of NK Cells

Having demonstrated that priming of cancer cells with PBP downregulates glucose uptake and metabolism and inhibits their proliferation, we next investigated whether the anti-cancer activity of NK cells was also influenced by cancer cell priming with PBP. Cell tracker pre-labeled B16F10 cells were pre-conditioned for 48 h in the presence or absence of either PBP or PBP-Scr peptide. At the end of the pre-conditioning interval, the B16F10 cells were treated for 4 h with NK cells and/or 150 μM PBP (or PBP-Scr), or co-treated with NK cells plus PBP or PBP-Scr, (see Figure 2A,C for the various treatment groups). The E:T (effector: target = NK:B16F10) cell ratio, was 5:1. As expected from the dose-response study (Figure 1A), PBP priming itself did not significantly increase cancer cell death. Although combining the addition of NK cells with PBP or with PBP priming both led to significant increases in cancer cell death, the maximal effect was achieved by preconditioning cancer cells with PBP followed by co-treatment with NK cells and PBP (Figure 2A). PBP-Scr priming of cancer cells failed to enhance cancer cell death by NK cells (Figure 2C).

Conceptually, the increased cell lethality after PBP priming of the cancer cells followed by the addition of NK cells might be the consequence of increased vulnerability of the cancer cells and/or increased anti-cancer efficacy of the NK cells. To explore the anti-cancer activity of the NK cells, we measured their cytokine secretion, including IFN-_γ_, TNF-α, and granzyme B. We observed that in fact NK cells secreted higher levels of the cytokines IFN-_γ_, TNF-α, and granzyme B upon engagement with PBP-primed cancer cells compared to non-PBP-primed cancer cells (Figure 2B). Just as the greater improvements in cytolytic activity of NK cells required PBP pre-conditioning of cancer cells, so did the secretion/production of cytokines. Co-treatment alone with PBP and NK cells was not sufficient to induce the cytokine-related phenotypic enhancements of NK cells, although co-treatment further augmented the effects of PBP pre-conditioning of cancer cells. Pretreatment or co-treatment of B16F10 cells with PBP-Scr had no effect whatsoever on cytokine secretion by NK cells (Figure 2D).

Along with the increased cytokine secretion, damage associated molecular pattern (DAMP) secretion was analyzed. HMGB1, serving as a representative of DAMP, in culture soup was measured at the same condition as Figure 2A. PBP priming did not increase the secretion of HMGB1 by cancer cells. However, when PBP-primed cancer cells were co-incubated with NK cells, there was greater secretion of HMGB1 than observed with non-primed cancer cells co-incubated with NK cells (Figure 3). Based on the data in conjunction with the inhibition in glucose metabolism induced by PBP priming (Figure 1C), we conjectured that PBP priming did not cause cancer cell death directly, but rather it contributed to the secretion of DAMP signals from cancer cells, leading to the alteration of TME to pro-immune environment. This mechanism enhances the anti-cancer activity of NK cells; dying cancer cells secrete more HMGB1, which could lead to synergistic effects on NK-mediated cytolytic activity against cancer cells.

To examine the possibility that PBP might directly modulate the anti-cancer activity of NK cells, we primed NK cells (instead of priming the B16F10 cancer cells) with PBP or PBP-Scr before their engagement with cancer cells (Appendix A). NK cells primed with PBP (or PBP-Scr) showed neither alterations in intracellular glucose concentration (Appendix A) nor enhancement of cancer cell cytolytic activity (Appendix A) or cytokine secretion (Appendix A). These data thus exclude a direct effect by PBP upon NK cells as the basis for their increased cancer cell cytolytic activity after PBP-priming of cancer cells. 

Taken together, the in vitro data presented in Section 3.1 and Section 3.2, above, establish that PBP priming of B16F10 cancer cells (but not of NK cells) alters their metabolism and proliferation. Importantly, they also suffer increased killing when exposed to NK cells, which in turn demonstrate enhanced cytokine secretion. We speculate that when cancer cells are primed with PBP, its binding to prohibitin impairs glucose metabolism, thereby inhibiting the Ras-C-Raf and Akt/PI3K pathways. Altered signaling modifies the TME (culture medium), shifting it from a pro-tumor to a pro-immune TME [12,31], triggering increased cytokine secretion, and other contributors to cancer cell killing, by NK cells.

### 3.3. PBP-Priming Enhances NK Cell-Mediated Inhibition of Tumor Growth

Having shown that tumor cell PBP priming enhances the anti-cancer activity of NK cells in vitro, we sought in vivo evidence that tumor priming with PBP would enhance the anti-cancer activity of adoptively transferred NK cells in obese mice with embedded B16F10 tumors, a model relevant to obese patients with cancer. The mice were assigned to four groups (*n* = 8/group); (1) control (CTRL), (2) PBP priming only (PBP), (3) NK only (NK) or (4) PBP priming + NK (PBP+NK). NK cells (Groups 3 and 4) or PBS (Groups 1 and 2) were injected directly into the tumors. PBP priming of tumors in groups 2 and 4 was with twice-weekly intraperitoneal injections of PBP, beginning on the day of NK cell or PBS injection.

Based on both tumoral luciferase activity (Figure 4A,B) and manual tumor size measurements (Figure 4C), the PBP+NK group (Group 4) achieved the greatest inhibition of tumor growth; either PBP or NK cells alone (Groups 2 and 3) was less efficacious. Interestingly, in mice with normal body weights, NK cells (in the absence of PBP priming) efficiently suppressed tumor growth; NK cells alone were much less effective in obese mice (Appendix A). These results imply that adoptively transferred NK cells lose cytolytic activity in the obese-associated pro-tumor TME. PBP-priming reinvigorates them. That there were no significant differences in body weight between the four groups is evidence that PBP was not toxic and that the treatments were well tolerated (Figure 4D and Appendix A).

Thus, the in vivo data obtained with the obese B16F10 tumor model are consistent with the cell culture studies; both the in vitro and the in vivo data support our hypothesis that pre-conditioning tumors with PBP restores or enhances cytolytic activity of NK cells added to tumor cell cultures or injected directly into tumors.

### 3.4. Tumor Priming with PBP Enhances In Vivo Cytokine Secretion by NK Cells

Earlier (see Section 3.2) we presented data that the increased in vitro cytolytic activity of NK cells against cancer cells primed with PBP was accompanied by increased cytokine secretion. To determine whether the improved tumor growth suppression of NK cells in vivo conferred by PBP treatment was likewise the consequence of enhanced cytokine secretion, cytokine content within tumors was measured by ELISA. The trends were similar to the in vitro cytokine secretion analysis (Figure 2). Although the levels of IFN-_γ_ (Figure 5A) and TNF-α (Figure 5B) were only modestly increased, granzyme B content (Figure 5C) was substantially higher in PBP-primed tumors than non-primed tumors. We note that when an NK cell encounters, recognizes, and seeks to destroy a cancer cell, the NK cell forms a synapse with the cancer cell. Within the NK cell, granzyme B is polarized toward the cancer cell, then secreted into the synaptic cleft in the form of secretory lysosomes; these migrate into the cancer cell [34,35]. Concomitantly, the NK cell secretes other cytokines (IFN-_γ_, TNF-α) into the surrounding environment; they bind to membrane receptors present on tumor cells to stimulate signaling pathways [35]. These two cytokine mechanisms converge to execute cancer cell killing [34,35,36]. Because IFN-_γ_ and TNF-α are largely extracellular, they are vulnerable to excessive loss during tissue sample processing for ELISA. Despite this limitation of our IFN-_γ_ and TNF-α assays, we can conclude that NK cells transferred to PBP-primed tumors in the obese mouse respond by enhanced cytokine secretion (Figure 5A–C), hence increased cytolytic activity (Figure 4A–C).

### 3.5. NK Cells Change Phenotype after Injection into In Vivo PBP-Primed Tumors

The findings of increased cytolytic activity (Figure 4) and enhanced cytokine secretion (Figure 5) by NK cells within PBP-primed tumors suggest the possibility of phenotypic change—specifically, invigoration—of NK cells injected into the tumors. To address this question, single-cell suspensions prepared from tumor tissue harvested after the in vivo experiments described earlier (see Section 3.3, Group 3—NK cells only; Group 4—PBP + NK cells) were analyzed by flow cytometry. Cells were stained for CD56, a marker of NK cell maturation, and NKG2A, PD-1, and inhibitory KIR (iKIR); the latter three are indicative of NK cell exhaustion [7]. By gating on human CD56-positive cells, the injected NK cell population was isolated from other cells in the single-cell suspension (compare Appendix A versus Appendix A). Then, the CD56-positive cells were further analyzed for receptor expression (Figure 6 and Appendix A). Importantly, NK cells isolated from PBP-primed tumors had a greater proportion of CD56^dim^ cells (Figure 6A) and a lower proportion of inhibitory receptor-positive cells (Figure 6B–D); both trends are emblematic of NK cells with greater cytotoxic activity, although the differences were not statistically significant except iKIR. The emergence in the PBP-primed tumors of a phenotypically-altered NK cell population indicative of highly active cytotoxic cells supports the hypothesis that priming cancer cells with PBP restores the cytolytic effector function of NK cells.

In addition to decreased metabolism of glucose by cancer cells and enhanced function of NK cells, PBP-priming might also alter the obese tissue microenvironment by blocking prohibitin on adipocytes; adipocytes, as well as melanoma cells, express sufficient prohibitin to render them targets for PBP binding. Inhibition of prohibitin in adipocytes interrupts the Akt/PI3K pathway, reducing both glucose uptake and metabolism [33]. The level of fructose, an intermediate metabolite produced by glucose metabolism, decreased in the PBP-treated adipocyte cell line, 3T3-L1, compared to the non-treatment group (Appendix A). This result indicates an additional effect of PBP, inhibition of glucose metabolism in fat cells within the tumor microenvironment. 

## 4. Discussion

Individuals who are overweight or obese, compared to those of normal or low weights, have an increased risk of developing cancer; those developing cancers are more likely to develop recurrence and have overall a less favorable prognosis [37,38]. When obese cancer patients are treated with adoptive immune cell transfer, the anti-cancer activity of the infused cells is rapidly suppressed in the TME [13], at least in part because increased glucose metabolism in the tumor serves to establish a TME especially supportive of tumor cells and no less hostile to immune cells. Immune cells, in general, are exhausted by their obesity-related accumulation of intracellular lipid and adipokines and by their continuous activation, a consequence of low-grade, chronic inflammation [5]. Consequently, inhibition of glucose metabolism within a tumor is asserted to be capable of switching a pro-tumor TME to a pro-immune TME and thus indirectly restoring cytolytic activities of immune cells.

In obesity, prohibitin is upregulated on the surface of cancer cells and adipocytes [18,22]. In cancer cells, prohibitin activates glucose metabolism by stimulating the Ras and Akt/PI3K pathways which modulate glucose/fructose uptake [18,25,26]. Prohibitin binding peptide (PBP), a nine amino acid peptide, is well established as an inhibitory ligand targeting prohibitin. In this study, we describe a new role for PBP—as an adjuvant that primes cancer cells, enabling them to restore or enhance the cytotoxic activity of NK cells (Figure 7). Predictably, PBP-primed cancer cells, versus non-primed ones, had lower concentrations of glucose and fructose, indicative of altered glucose metabolism; inhibiting glucose/fructose uptake in cancer cells in turn modulates the TME [11,12]. Based on this finding, we pre-conditioned cancer cells in vitro with PBP to examine consequent changes in the cytotoxic activity of NK cells. PBP had no significant cytotoxic effects on either cancer cells or NK cells in isolation; however, priming cancer cells with PBP notably enhanced the cytotoxic activity of and cytokine secretion by NK cells, and DAMP secretion by cancer cells when co-incubated with NK cells. We highlight that PBP weakens cancer cells against NK cells and generates pro-immune environment to activate NK cells, in which the NK cells eradicate cancer cells more effectively in response to the increase in DAMP secretion. It would be a cascade effect triggered by PBP, which can be an adjuvant to adoptive immunotherapy.

To mimic the clinical scenario of adoptive immunotherapy for obese-cancer patients, human NK-92 cells were adoptively transferred by direct injection into tumors in an obese mouse model. We reasoned that intra-tumoral injection would best reveal the influence of PBP-priming of the tumor on the activity of NK cells. Tumor-bearing mice with normal body weight were also administered NK cells but not treated with PBP. This allowed a comparison of the therapeutic efficacy of adoptively transferred NK cells between obese mice and those of normal weight. Notably, a significant difference in tumor growth between obese and normal mice was observed; in the absence of PBP injection, the efficacy of adoptive NK cell transfer in obese mice was lower than that in normal mice. Interestingly, the inhibition of tumor growth in obese mice treated with PBP-priming plus NK cell injection approximated that of NK cells alone in mice of normal weight. In addition, obese mice with PBP injection presented the reduced tumor growth compared to the control group. The result might be due to the PBP-mediated alteration in TME caused by the changes in glucose metabolism and the increased DAMP secretion, both of which could activate autologous mouse NK cells. This observation adds strong support to our premise that PBP constitutes an effective adjuvant to restore cytolytic activity of adoptively transferred NK cells in obese patients.

PBP-priming of the tumor increased cytokine secretion by NK cells (Figure 5), increasing their cytolytic anti-cancer activity, which was substantiated by two measures of tumor growth (Figure 4). Evidence that both manifestations of PBP-tumor priming were the consequence of restoring function to NK cells was sought by flow cytometry of NK cells extracted from the tumors. Normally, CD56^dim^ and CD16-positivity are emblematic of strong effector NK cells while CD56^bright^ and CD16-negative NK cells are characteristically weakly cytotoxic [39,40,41]. Recent data, however, indicate that CD56^dim^ and CD16-negative NK cells exert moderate cytotoxic activity that is nonetheless greater than that found with CD56^bright^ and CD16-positive/-negative NK cells [39,42]. This finding emphasizes that CD56^dim^ is the determinative marker of effector function of NK cells. Moreover, the frequency of CD56^dim^ NK-92 cells had a positive correlation with cytotoxicity and cytokine secretion when exposed to breast cancer cells [43]. The NK-92 cells used in our study were initially either CD16-negative or -low; subsequently, CD16 expression levels were not different between PBP-primed and PBP non-primed groups (data not shown). Because NK cells showing CD56^dim^ regardless of CD16 status have strong cytotoxic activity, we gated cells into CD56^dim^ and CD56^bright^ subsets for comparison with their cytolytic activities. We found that the CD56^dim^ NK cell population was larger in PBP-primed tumors than in tumors not primed. This population—representing strong cytolytic activity—is critical to the enhanced anti-cancer activity of NK cells because of their increased cytokine release.

Along with enriched CD56^dim^ NK cells in PBP-primed tumors, we noted that expression levels of inhibitory antigens on the NK cells isolated from PBP-primed tumors were lower versus non-primed tumors (Figure 6). Exhausted NK cells and NK cells in TME tend to have high expression levels of inhibitory antigens; their cytolytic activities are suppressed via binding of inhibitory antigens to receptors on cancer cells [44,45]. Lower expression levels of inhibitory antigens and therefore reduced interactions with cancer cell receptors would help maintain NK cells in their active state. There are efforts to reinvigorate anti-cancer activity of NK cells with blocking antibodies against inhibitory antigens on NK cells; this represents another immune checkpoint blockade drug for immune-cancer therapy [45,46]. These coincident characteristics—CD56^dim^ and low expression level of inhibitory antigens- of the NK cells collected from PBP-primed tumors are known to be associated with enriched production/secretion of cytokines. Blocking of inhibitory antigens with an antibody stimulates NK cells to increase production of cytokines, including IFN-_γ_, TNF-α, perforin, and granzyme B [47,48]. We demonstrated both in vitro and in vivo that enhanced secretion of cytokines by NK cells followed engagement with cancer cells pre-conditioned with PBP (Figure 2 and Figure 5). Both sets of observations are consonant with improvement of NK cell activity through down-regulation of inhibitory antigens and CD56 (to the CD56^dim^ state). However, we emphasize that PBP enhances the cytotoxic activity of NK cells by priming the cancer cell and thus indirectly affecting the TME, not by any direct effect on the NK cells.

In both cancer cells and adipocytes, prohibitin modulates the Akt/PI3K pathway [33]. We have found that in addition to the effects of PBP on cancer cells, it also inhibits glucose metabolism of adipocytes, demonstrated by decreased fructose concentration reflecting altered glucose/fructose uptake and/or metabolism (Appendix A). Reduced glucose metabolism is accompanied by decreased adipogenesis and down-regulated biological activity; both reduce adipokine secretion and alter the adipose tissue microenvironment [8,10,49] Because of the role of increased adiposity and adipokines in the activation of PI3K and RAS signaling in tumors of overweight or obese cancer patients, clinical trials have targeted adiposity as a therapeutic strategy [50]. PBP-priming of cancer cells, which counters the deleterious interactions between adipose tissue and tumor in the setting of obesity, renders the tumor and adipose tissue microenvironments more favorable to endogenous and exogenous immune cells, facilitating immunotherapies.

## 5. Conclusions

In both in vitro and in vivo studies, we show that the PBP, by inhibiting prohibitin overexpressed on cancer cells, has direct effects on the cancer cell and indirect effects—and possibly direct ones—on the TME. As a consequence, adoptively transferred NK cells are rejuvenated, demonstrating increased cytokine secretion and tumor cell killing (Figure 7). In the setting of obesity, PBP dampens the upregulated glucose metabolism in the tumor, shifting the TME to a pro-immune TME. In this modified TME, NK cells down-regulate both CD56 and inhibitory receptor expression. In our mouse model, the resulting NK cells with a stronger cytolytic phenotype increase production and secretion of cytokines, enhancing their cytolytic activity, thereby becoming important effector cells for the eradication of the tumor. Although our in vitro experiment is designed to mimic the TME in obese tumor, limitations exist; however, we were able to demonstrate that PBP reduces glucose metabolism in cancer cells. Further mechanistic studies are needed to more precisely define the relevant changes within the TME or cancer cells caused by PBP-priming and the means by which these factors dictate the fate of adoptively transferred NK cells in the TME. That said, our data suggest a new role of the PBP as an adjuvant to immunotherapy in obese patients with cancer.

## Figures and Tables

**Figure 1 pharmaceutics-13-01279-f001:**
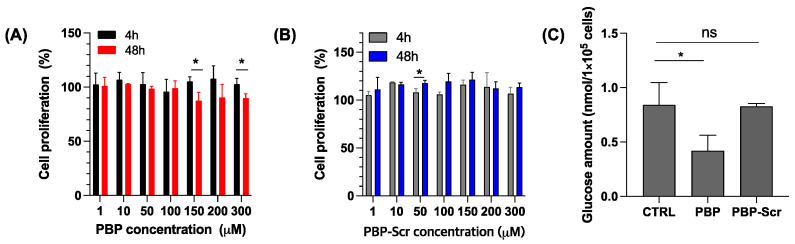
Effects of PBP or PBP-Scramble (PBP-Scr) priming on cancer cell viability and intracellular glucose. (**A**,**B**) B16F10 cancer cell viability after treatment with increasing concentrations of PBP or PBP-Scr for four hours or 48 h. Statistical analysis was performed using a *t*-test, *p* < 0.05. (**C**) Glucose amount in B16F10 cell lysates after PBP or PBP Scr (150 µM) treatment for 48 h compared to untreated cells. Statistical analysis of the data was with two-way ANOVA, Tukey test (* *p* < 0.03, and ns = non-significance).

**Figure 2 pharmaceutics-13-01279-f002:**
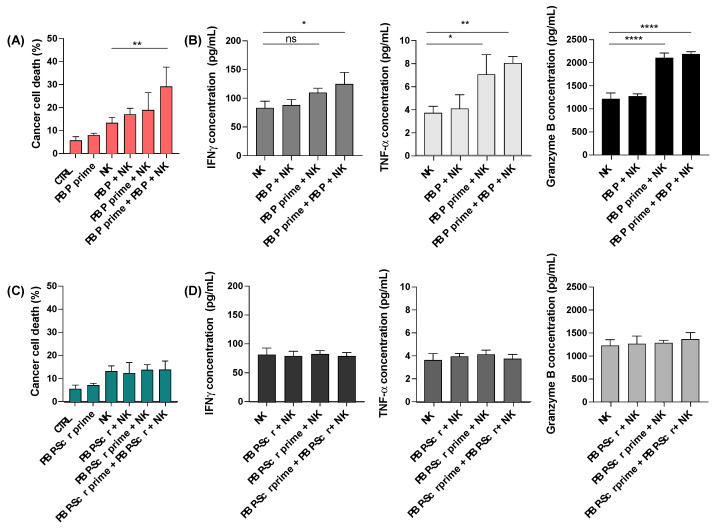
PBP or PBP-Scr effects on anti-cancer activity of NK cells. Enhanced anti-cancer activity of NK cells after PBP priming of cancer cells is accompanied by increased cytokine secretion by NK cells. (**A**,**C**) Cytotoxic activity of NK cells against B16F10 cancer cells with or without 48h PBP or PBP-Scr priming (“PBP prime” or “PBP-Scr prime”) and/or co-treatment (“PBP” or “PBP-Scr”). (**B**,**D**) Cytokine (IFN-_γ_, TNF-α, and granzyme B) concentrations determined by ELISA in the culture supernatants from (**A**,**C**), respectively. One-way ANOVA, Tukey test was performed for statistical analysis (* *p* < 0.03, ** *p* < 0.002, **** *p* < 0.0001, and ns = non-significance) and *n* = 3.

**Figure 3 pharmaceutics-13-01279-f003:**
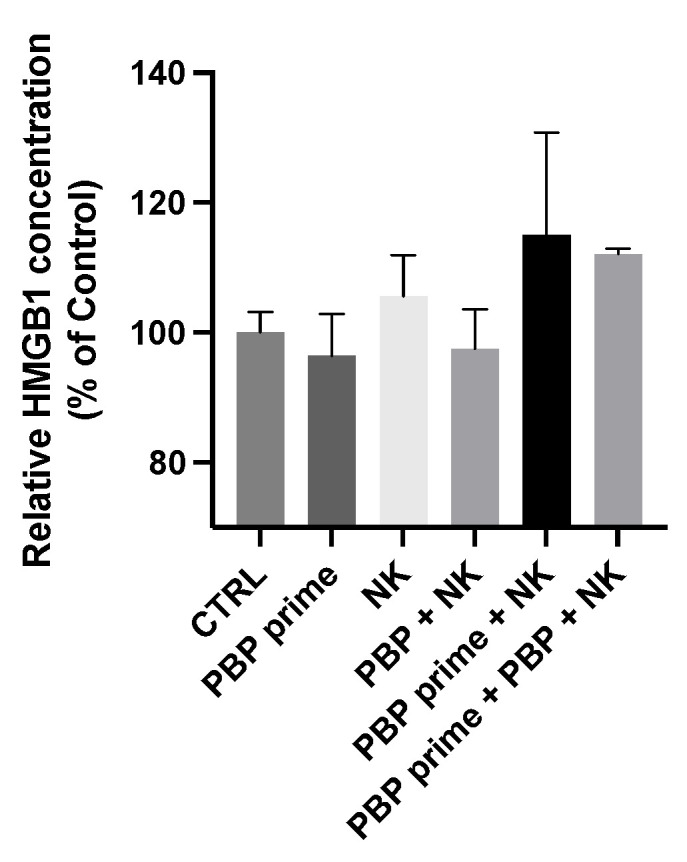
Relative HMGB1 concentration in culture supernatants. HMGB1 secretion by PBP-primed B16F10 cells did not vary significantly from control cultures. However, PBP primed B16F10 cells cultured with NK cells exhibited greater HMGB1 secretion into the culture supernatants.

**Figure 4 pharmaceutics-13-01279-f004:**
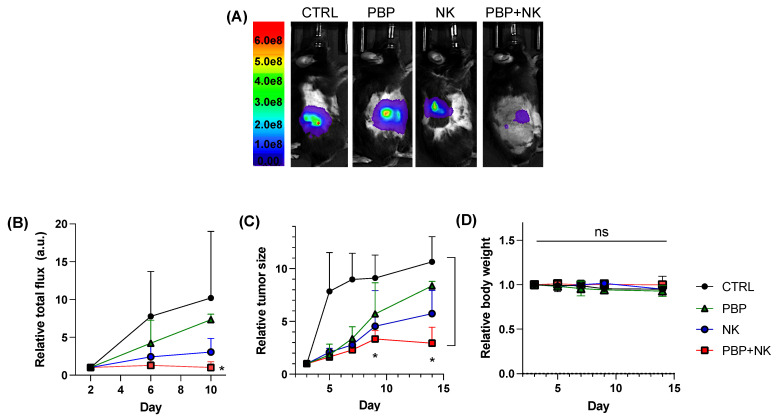
In vivo anti-tumor activity of NK cells and/or PBP. (**A**) In vivo luciferase bioluminescence imaging on day 10. (**B**) Relative luciferase flux. (**C**) Relative tumor size based on manual measurements and (**D**) relative body weight during treatment. Two-way ANOVA, Tukey test, *p* < 0.05 was conducted for statistical analysis (* *p* < 0.03, and ns = non-significance); *n* = 8.

**Figure 5 pharmaceutics-13-01279-f005:**
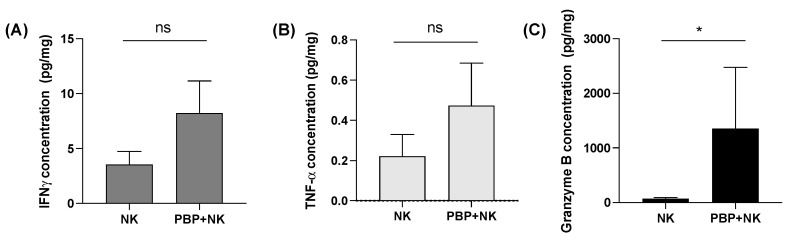
Cytokine levels determined by ELISA of tumor lysates: (**A**) IFN-γ, (**B**) TNF-α, and (**C**) Granzyme B. One-way ANOVA, Tukey test, *p* < 0.05 was performed for statistical analysis (* *p* < 0.03, and ns = non-significance).

**Figure 6 pharmaceutics-13-01279-f006:**
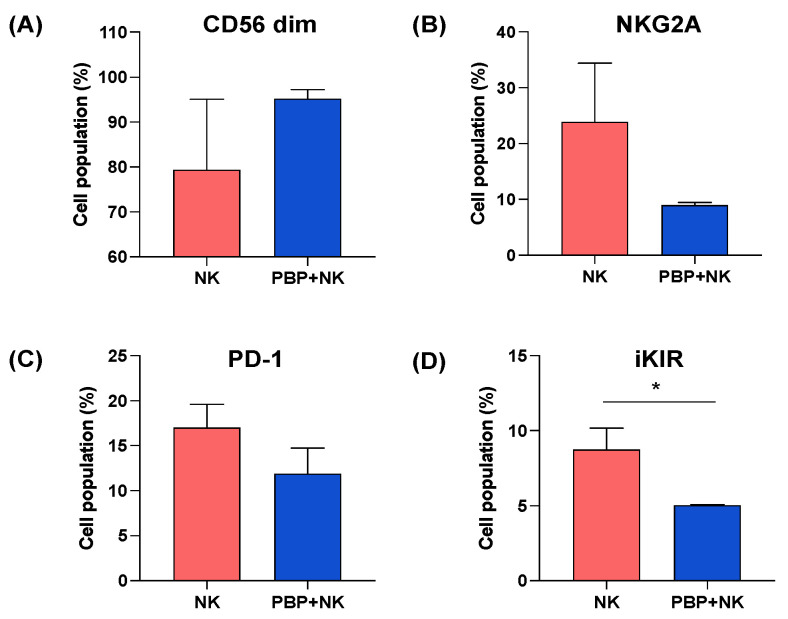
Phenotyping of NK cells isolated from tumor tissues: (**A**) population of CD56^dim^ NK cells, (**B**–**D**) populations expressing inhibitory receptors: NKG2A, PD-1, and inhibitory KIR (iKIR). NK cells were analyzed by gating CD56 positive cells from the total cell population. The bar graphs indicate the proportion of cells in the populations defined by the selection bars on the histograms shown in Appendix A. Statistical analysis was performed using the *t*-test, * *p* < 0.05.

**Figure 7 pharmaceutics-13-01279-f007:**
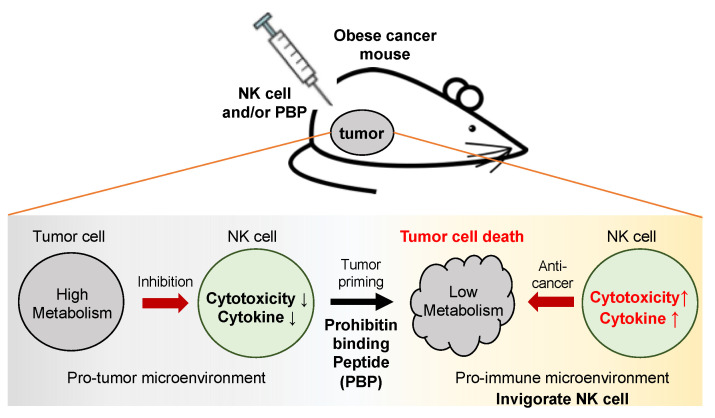
Schematic illustration. The tumors of obese cancer mice are directly injected with NK cells with or without treatment with prohibitin binding peptide (PBP). PBP priming inhibits glucose metabolism in cancer cells and reduces the production of metabolites, leading to changes in the tumor microenvironment rendering it more supportive of functional immune cells. Finally, NK cells regain cytotoxic activity, inhibiting tumor growth more efficiently in PBP-primed cancers in obese mice.

## Data Availability

The data presented in this study are available in this article: Peptide Adjuvant to invigorate Cytolytic Activity of NK cells in an Obese Mouse Cancer Model.

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
