# Peer review of "Peptide Adjuvant to Invigorate Cytolytic Activity of NK Cells in an Obese Mouse Cancer Model"

_pharmaceutics, 2021, doi:10.3390/pharmaceutics13081279_

Round 1
Reviewer 1 Report
The authors have addressed my concerns seriously and performed the necessary experiments rigorously. I am satisfied with their approach and most of their explanation/interpretation. However, in the spirit of ensuring thoroughness and coherent logic, I would like to clarify the following concerning the potential contribution of DAMPs:
“As shown in Figure R4, PBP priming did not increase the secretion of HMGB1 by B16F10 cells. However, when PBP-primed cancer cells were incubated with NK cells, there was greater secretion of HMGB1 than observed with non-primed cancer cells plus NK cells. Based on these data in conjunction with glucose metabolism inhibition induced by PBP priming (Figure R1), we speculate that PBP priming does not render cancer cell apoptotic or foster HMGB1 secretion, but rather it contributes to a pro-immune environment.” (p.6 of rebuttal letter)
If PBP priming increased HMGB1 secretion in the presence of NK cells, and HMGB1 is a DAMP that promotes immune response, then would that not be at least a potential secondary means through which PBP achieves its anti-tumor effects in vivo? Could the authors please explain the insistence that PBP only acts via altered by intracellular glucose metabolism? After all both the glucose and the HMGB1 data are newly presented and where necessary, the authors should be willing to accommodate new information. This especially so given the strong contrast between the modest effects of PBP on intracellular glucose in vitro (new Fig. 1B) and the dramatic anti-tumor activities in Fig. 3C.
In the same vein, the authors stated that:
“The experiments we have performed in response to Reviewer 1’s earlier criticisms have demonstrated that PBP priming impairs glucose metabolism and does not elicit DAMPs. These results lend strong support to our hypothesis that PBP-priming does not directly cause cancer cell death but instead it triggers inhibition of cancer cell proliferation and induces alterations of TME through suppression of cancer cell glucose metabolism.” (p.7 of rebuttal letter)
The tumor microenvironment (TME) is an in vivo multi-tissue context and since the authors have presented evidence that HMGB1 production is indeed increased when primed cancer cells are co-cultured with NK cells, then it is not logical to be emphasising that PBP does not induce B16F10 cells to produce DAMPs in vitro. In fact, this is precisely the point of my earlier comment:
d) Along the same lines, the authors should address the distinct possibility that PBP could trigger tumor cell death in vivo to elicit more vigorous NK activities independent of the prohibitin-glucose metabolism axis;
The authors are at liberty to distinguish primary cell autonomous effects (i.e. PBP/glucose axis) measured in vitro and secondary non-cell autonomous effects (through the induction of HMGB1). However, it would be a disservice to this manuscript to miss the point that PBP could trigger a compounding of PBP and DAMP effects in the TME. I do not believe additional experiments are necessary, but the authors should address this in their discussion of their data.
Author Response
We appreciate the reviewer’s comments and advice. We agree with the reviewer that the increase in HMGB1-caused by PBP would bring a secondary means to the activation of NK cells or alteration in TME along with the inhibition in glucose metabolism. PBP appears to be involved with other complicated mechanisms underlying the enhanced anti-cancer effects of NK cells. We are currently conducting studies to define more detailed mechanisms. The HMGB1 data are included in the Results section part 3.2 (Figure 3), and is discussed in lines 437 - 453 according to the reviewer’s suggestion. Our future plans to investigate a deeper mechanism of PBP are described in the conclusion line 527. The newly inserted part is colored in red letters.

Reviewer 2 Report
The manuscript of Han et al describes the augmentation of NK cell anti-tumour function with the use of prohibitin-binding peptide. The data appear sound and are presented clearly. However, while the in vivo effects of the NK+PBP is quite striking, the in vitro to support this show a very modest effect of PBP on NK cells killing. This leads me to think that the mechanism of is more complicated or has been missed. Have the authors got other data and/or can they speculate further on the underlying mechanism of enhanced NK anti-tumour function? Is it possibly that other factors that support tumour growth (perhaps generated from adipose tissue) and NK cell resistance in vivo that are not sufficiently modellable in vitro?
I have the other suggestions/comments:
All PBP-Scr priming data is shown as supplementary figures – This data should realistically be contained within the same figure as the other main data.
Does PBP have an effect on NK cell viability or proliferation?
For the data shown in Figure 5, was MFI analysed and was this data significantly different. Also, it should be stated in the main text that the differences observed were not statically significant.
Author Response
We appreciate the reviewer’s detailed comments. We agree with the reviewer that PBP might have more complicated mechanisms beyond the inhibition of glucose metabolism in cancer cells, which in turn lead to the activation of NK cells. We speculate that PBP might have an effect on not only cancer cells but also adipocytes based on the data, Figure SI8. However, we were skeptical about the PBP effect on NK cells because we did not see any direct effects of PBP on the enhancement in anti-cancer effects of NK cells from the in vitro data (Figure SI4). In addition, as the reviewer mentioned, in vivo obese model might have other factors to support tumor growth and NK cell resistance that are not sufficiently modellable in vitro. That is a limitation of our in vitro data to the discovery of potential other mechanisms. However, our in vitro data is sufficient enough to demonstrate that PBP has effects on the inhibition in glucose metabolism in cancer cells, and the PBP-mediated glucose metabolism inhibition is associated with the increase in anti-cancer activity of NK cells. For the detailed correlation between PBP and cancer or adipocytes, we are planning to investigate the deeper mechanism in our next study (lines 525-530).
We have added PBP-Scr priming data in the main body of the manuscript (Figures 1 and 2), while the fructose data of PBP is moved to the supporting materials in order to match the data set.
We have not tested the effects of PBP on NK cell viability or proliferation. However, we assure that PBP would not have any effects on NK cells because PBP did not affect the glucose metabolism and the cytokine secretion profile of NK cells (Figure SI4).
As following reviewer’s suggestion, we have stated that the differences were not statistically significant in Figure 5 (which is now figure 6), Result line 384. The revised part is marked as red-colored letter.

This manuscript is a resubmission of an earlier submission. The following is a list of the peer review reports and author responses from that submission.
Round 1
Reviewer 1 Report
In their manuscript entitled “Peptide Adjuvant to Invigorate Cytolytic Activity of NK cells in an Obese Mouse Cancer Model”, Han and co-workers tackle an important question limiting current immuno cancer therapy, namely the exhaustion of anti-tumor NK cells in obese individuals due to metabolic abnormalities in the tumor microenvironment. The authors sought to modulate the metabolism of cancer cells through the use of an interfering peptide against prohibitin (called prohibitin-binding peptide or PBP). The over-expression of prohibitin was previously shown to elevate glucose metabolism and consequently fashions a pro-tumorigenic tumor microenvironment. They observed that PBP had only modest anti-proliferative activity in B16F10 melanoma cells, but pre-incubation with PBP appears to augment the cytolytic activity of cultured NK cells. This was accompanied by increased IFN-gamma, TNF-alpha and Granzyme B release. The authors then tested the efficacy of PBP in vivo by transplanting B16F10 cells subcutaneously into obese mice and combining PBP injection and adoptive transfer of NK cells. The authors report that the combined treatment of PBP and NK cells resulted in significant reduction of tumor cell-specific luciferase flux over time, which was mimicked by changes in tumor size. Lastly, the authors detected significant increases in granzyme B release as a result of PBP and NK cell co-treatment, though not in IFN-gamma and TNF-alpha secretion.
This manuscript is very well constructed and written. The logical flow, especially in the Materials and Methods section, is fluent; and the purpose and significance of the study is clearly explained. Although handicapped by several deficiencies, the experiments appear well conducted.
The strength of this manuscript is that its subject matter is important and topical, and its experiments well conducted. The major drawback of this study is the hasty interpretation of the data that led to the authors overlooking some important issues (see below). Overall, the conclusion of this study is not sufficiently supported by the data presented, many of which are quite subtle. This study could be strengthened by addressing the following points:
1) The prohibitin-binding peptide (PBP)
- The authors have previously reported that this peptide was used to target prohibitin in adipocytes (Won et al; ref 29). Others (Kolonin et al.; ref 27) have done likewise. In this paper, the authors argue that the target of PBP is the prohibitin in the melanoma cells. Was there something different in the flanking sequence of the short anti-prohibitin CKGGRAKDC region? Does this peptide sequence target mouse prohibitin? This part should be clarified in the Materials and Methods, especially in view of the data from 3T3-L1 cells, and elsewhere in the text;
- Many of the authors’ observations could potentially be explained by a contamination with a toll-like receptor agonist (or agonists). This was not ruled out. The authors should conduct a pyrogen (e.g. LPS) test to confirm that the PBP preparation is free of contaminants.
2) The notion of metabolic alteration of tumor cells
- This is a central thesis of this manuscript. Yet, the sole evidence of PBP achieving the stated effect of metabolic alteration of B16F10 cells is the measurement of intracellular fructose (as a surrogate for intracellular glucose). For a study that set out to investigate the concept of altering cancer cell glucose metabolism via inhibition of prohibitin, it is not clear why the authors did not simply measure changes in culture supernatant glucose. This is a glaring omission. The authors should perform a comprehensive suite of metabolic measurement to quantify the in vitro and in vivo changes in cellular metabolism in response to PBP;
- In its place, the authors measured the secretion of two cytokines - IFN-gamma and TNF-alpha in response to PBP treatment. The amounts of cytokines detected by ELISAs seem negligible. This was especially so for TNF-alpha, which was detected at sub-picogram levels, and thus likely to be assay backgrounds.
3) Interpretation of data
- A shortcoming of the current study is the interpretation of data. Essentially, the authors are trying to fit their in vitro and in vivo observations without taking into sufficient consideration the vastly different contexts. There are important questions that the authors have to address:
a) Was the reduction of B16F10 cell viability in vitro really due to PBP inhibiting prohibitin? Is prohibitin over-expressed in B16F10 cells? Would RNAi targeting of prohibitin achieve a similar reduction in cell viability? Would PBP still have an effect in RNAi knockdown cells?
b) Was the reduction of B16F10 viability accompanied by increased apoptosis? If so, the authors should address the possibility that increased NK-mediated killing could be a function of increased DAMPs (damage associated molecular pattern) compounds in the culture supernatant. This possibility should be experimentally excluded;
c) The selection of dose - the authors based their selection of 150µM PBP as the appropriate dose based on their findings presented in Fig 1A. Given that PEP could cause appreciable drops in cell viability within 4h, would it not make better sense to use a dose that DID NOT cause such strong cytostatic/cytotoxic activities? This would favour the author's argument that the reduction in tumor growth was due to increased NK-mediated anti-tumor activities.
d) Along the same lines, the authors should address the distinct possibility that PBP could trigger tumor cell death in vivo to elicit more vigorous NK activities independent of the prohibitin-glucose metabolism axis;
e) Many of the effects are modest or subtle. The authors should make a special effort to annotate where the differences are insignificant, esp. Fig. 3C; but also Fig 1A; Fig 1B; Fig. 2B and Fig. 4B;
Minor points:
- As far as I could see, the source and nature of the adoptively transferred NK cells was not described;
- Many of the bar graphs have formatting artefacts:
• Greek fonts are not properly displayed on screen and in print (e.g. y-axis labels of Fig. 2A-B);
• font and spacing of x-axis labels in Fig. 1C, 2A-C are problematic;
• Fig 3A should be enlarged;
• Fig 3B-D would benefit from graph legends.
Reviewer 2 Report
The manuscript by Han and colleagues, entitled ‘Peptide Adjuvant to Invigorate Cytolytic Activity of NK cells in an Obese Mouse Cancer Model’ reports that inhibiting the prohibitin signaling pathway reduces glucose metabolism in obese cancer cells can restore NK the cytotoxic activity of NK cells. The author performed the assays to prime tumor cells with an inhibitory prohibitin-binding peptide (PBP). After tumor priming, the activities of adoptively transferred NK cells can be enhanced. The results suggest that PBP has the potential as an adjuvant to enhance the activities of adoptively transferred NK cells in cancer patients with obesity.
The “results” session of manuscript is divided into four parts as follows:
#1: Tumor priming with PBP enhances in vitro anti-cancer efficacy of NK cells
#2: In vitro tumor priming with PBP enhance cytokine secretion of NK cells
#3: PBP-priming enhancing NK cell-mediated inhibition of tumor growth
#4: Tumor priming with PBP enhances in vivo cytolytic activity of NK cells
#5: In vivo NK cells change phenotype on PBP-primed tumor group.
The novelty of this manuscript is that the authors proposed a novel hypothesis, which include “inhibiting the prohibitin signaling pathway reduces glucose metabolism in obese cancer cells can restore NK the cytotoxic activity of NK cells”.
However, a few concerns are detailed below.
Major Points:
- The potential molecular mechanisms of NK cells with enhanced anti-tumor activities upon stimulation upon PBP priming on tumor cells are not clear.
- The possible correlation between NK cell activities (e.g., Cytokine production and killing activities) and PBP priming can be further analyzed.
Minor Points:
- In Figure 1A, statistical analysis is missing.
- In Figure 1A, a control peptide is required to interpret the data.
- In Figure 1C, preconditioning of the B16F10 cancer cells with a control peptide is required to interpret the data.
- In Figure 2, preconditioning of the B16F10 cancer cells with a control peptide, then measuring NK cell cytokine production is essential.
- In Figure 3, preconditioning of the B16F10 cancer cells with a control peptide, then measuring NK cell anti-tumor activity is essential.
- In Figure 3, the genetical background of obese B16F10 tumor model is not clear.
- In Figure 4, statistical analysis is missing.
- In Figure 5, other major NK cell activating receptors and inhibitory receptors are required. Again, a control peptide group is essential to properly interpret the data.

Reviewer 3 Report
Dear Authors,
The article “Peptide Adjuvant to Invigorate Cytolytic Activity of NK cells in an Obese Mouse Cancer Model” is very interesting work, reporting the study connected with a new role for PBP - as an adjuvant which primes cancer cells to restore or enhance the cytotoxic activity of exhausted NK cells.
The article is quite well-written with proper study design and reports interest findings. I recommend its publication, although before it I encourage Authors to revise some minor issues listed below:
- English is not always suitable and it is suggested to be revised by a native speaker. Some examples, but not limited to, are:
- line 20: “Adoptive immunotherapy has emerges” it should be “has emerged”
- percentage symbol, that accompany numbers always should be closed up with the numerals (please refer to the entire manuscript)
- please put spaces between symbols like "=" or "<" throughout the manuscript and under the figure description.
- Other concerns:
- line 139: “dose 150 μM/kg 139 in 100 μl PBS” I think that it should be μg/kg.
- Please unify the unit style: according to International Bureau of Weights and Measures more correct is the version with space after the number, e.g. 4 μM.
- line 164: typo mistake: “ANOVA with turkey tests” it should be “ANOVA with Tukey test”
- all the figures: please put the info regarding the statistics on each figure (statistical significance)
- Figures 2 and 4 have captions with very small size. Please enlarge figures, for example occupying all page width, to make them more readable and attractive to the reader. Figures 1 and 3 are more readable but a small enlargement will also be welcomed.
- For every kit, reagent and chemical cited in Material and methods, authors need to detail the name of the vendor, the city of the vendor and its country. In the case of USA cities, also the State.. Example: Sigma-Aldrich Merck, Sant Louis, MO, USA. The way of citation is irregular: some are missed, others with the company, others with the State but not the city, and so on. Please homogenize as requested and please also do it also for antibiotics as penicillin and streptomycin.
- Please revise carefully the manuscript to find possible typos. For example "interlukin" (line 91) should be "interleukin".
Kind regards